# Novel data show expert wildlife agencies are important to endangered species protection

Michael J. Evans[1], Jacob W. Malcom[1,2] & Ya-Wei Li[3]

To protect biodiversity, conservation laws should be evaluated and improved using data. We provide a comprehensive assessment of how a key provision of the U.S. Endangered Species Act (ESA) is implemented: consultation to ensure federal actions do not jeopardize the existence of listed species. Data from all 24,893 consultations recorded by the National Marine Fisheries Service (NMFS) from 2000–2017 show federal agencies and NMFS frequently agreed (79%) on how federal actions would affect listed species. In cases of disagreement, agencies most often (71%) underestimated effects relative to the conclusions of species experts at NMFS. Such instances can have deleterious consequences for imperiled species. In 22 consultations covering 14 species, agencies concluded that an action would not harm species while NMFS determined the action would jeopardize species' existence. These results affirm the importance of the role of NMFS in preventing federal actions from jeopardizing listed species. Excluding expert agencies from consultation compromises biodiversity conservation, but we identify approaches that improve consultation efficiency without sacrificing species protections.

[1] Center for Conservation Innovation, Defenders of Wildlife, 1130 17th St, NW, Washington, DC 20036, USA. [2] Department of Environmental Science and Policy, George Mason University, 4400 University Dr., Fairfax, VA 22030, USA. [3] Environmental Policy Innovation Center, 777 6th St, NW, Washington, DC 20001, USA. Correspondence and requests for materials should be addressed to M.J.E. (email: mevans@defenders.org)

Data-driven decision-making is an important and growing theme in society. In governance, data can provide an accurate picture of how laws and policies are implemented, highlight real successes, and identify shortcomings[1,2]. Data may be of particular value to governance decisions that are framed as a choice between polarized extremes, such as the role of regulation in society[3]. Data can temper extreme rhetoric by providing a more nuanced evaluation of regulatory approaches[4], offering evidence for competing alternatives, and identifying areas for compromise[5]. Although data availability does not guarantee their use in decision-making[6], data collection and analysis are the first steps to realizing these benefits. There is a pressing need to use data to inform environmental policy because it often involves opposing ideals, with biodiversity protection often (unnecessarily) pitted against economic development. In the U.S., a highly-polarized political climate has catalyzed an unprecedented number of legislative proposals undermining conservation laws, often based on the claim that conservation hinders economic growth. These conflicting views raise an important question: How can conservation laws be most effective and cost-efficient to implement?

For decades, government agencies, politicians, and the public have offered competing approaches to balancing economic interests with species conservation[7,8]. One important but controversial approach is to empower dedicated agencies to evaluate and limit proposed activities that harm imperiled species. The U.S. Endangered Species Act (ESA) is among the world's strongest biodiversity conservation laws—in large part because it vests expert agencies with oversight authority. Section 7 of the ESA requires that federal agencies consult with the U.S. Fish and Wildlife Service (FWS) or National Marine Fisheries Service (NMFS; together "the Services") to ensure that actions the agencies take, fund, or permit will not jeopardize the existence of any species on the endangered species list or adversely modify these species' critical habitat. All listed species including any Distinct Population Segments (DPS) or Evolutionarily Significant Units (ESU)—distinct segments of a species that can be independently listed under the ESA[9]—require consultation[10].

Federal agencies can determine on their own authority that a proposed action will have 'no effect' on listed species, in which case the action proceeds without involvement of the Services. If an agency determines a proposed action 'may affect' listed species, then either FWS or NMFS reviews whether the proposed action is "likely to adversely affect" (LAA) a species or critical habitat. This informal consultation ends if the Services determine that the proposed action will have 'no effect' or may affect but is "not likely to adversely affect" (NLAA) listed species or critical habitat. If the Services make an LAA determination, formal consultation is initiated. The Services then evaluate whether the proposed action will jeopardize species (i.e., appreciably reduce the species' probability of survival) or destroy/adversely modify their critical habitat (See Supplementary Fig. 1 for a diagram of possible consultation outcomes). If either of these outcomes is likely, the Services must suggest "reasonable and prudent alternatives" that agencies can implement to reduce, or offset harm caused by the proposed action. If no alternatives are available, the action cannot proceed without violating the ESA (unless exempted by the Cabinet-level Endangered Species Act Committee). We hereafter refer to federal agencies engaged in the consultation process as action agencies, and actions requiring consultation as proposed actions. Consultations may cover multiple species, each of which may be affected differently by a proposed action. We refer to the individual species effects of an action as a determination. Thus, many consultations have multiple determinations.

The consultation requirement is important for biodiversity conservation under the ESA because the Services' have unique expertise vital to accurately identifying the effects of proposed actions on species' conservation prospects. The Services alone have a legal mandate to determine how actions affect listed species, lead the development of plans to recover listed species, and periodically assesses their conservation status. Consultation is also the primary regulatory protection in the ESA for plants[11]. Simultaneously, consultation is criticized as inefficient and burdensome by some parties, although recent research indicates this criticism is often overstated[12]. Nonetheless, allowing action agencies to determine the effects of their actions on imperiled species themselves, without the input of the Services, continues to be proposed. These self-consultation approaches appeared in several U.S. legislative and administrative decisions including the 2004 pesticide counterpart rule[13], the National Forest counterpart rule[14], and alternative consultation regulations during the G.W. Bush administration[15]. Interest in reducing expert agency involvement continues to this day: exemptions from consultation with the Services for forest management and pesticide registrations were proposed in a U.S. House of Representatives draft of the 2018 Farm Bill[16]. A critical, outstanding question is whether self-consultation alternatives effectively conserve species protected by the ESA or simply alleviate conservation obligations. This question has never been quantitatively evaluated despite the controversy surrounding this issue, because the data have not previously been available.

We provide the first data-driven examination of the consultation program of NMFS, the expert agency responsible for evaluating federal actions occurring in marine environments or affecting most anadromous fishes. Our goal was to answer two fundamental questions. First, how often do federal agencies and NMFS disagree about the effects of their actions on threatened and endangered species? This knowledge is important to assess whether self-consultation would maintain or reduce protections for listed species. Second, what are the general patterns of consultation outcomes, including jeopardy and adverse modification determinations? While we could not evaluate the effect of consultation on species status, these questions are critical for understanding the conservation impacts of a consultation program and predicting the effects of proposed changes to the law.

We show that over the past 17 years, more than one-fifth of consultations have included proposed determinations from action agencies that could result in the under- or over-protection of species relative to the expert NMFS determinations. Certain agencies and types of actions are more likely to be at odds with the NMFS conclusion, and the results strongly indicate that in general NMFS's expertise on species biology and threats is critical for protecting threatened and endangered species while self-consultation could compromise this purpose of the ESA. Second, NMFS rarely determined that federal actions would jeopardize species or adversely modify critical habitat, and no actions were stopped because of NMFS finding jeopardy or adverse modification without reasonable and prudent alternatives. Together with quantitative and qualitative descriptions of consultations, such as patterns of jeopardy determinations, the data point to strengths of the section 7 program and can be used to identify potential improvements in its implementation.

## Results

**Consultation patterns and trends**. The Public Consultation Tracking System (PCTS) database shows that NMFS biologists recorded 19,826 informal and 4934 formal consultations (19.9% formal) from January 2000 through June 2017. These numbers exclude consultations recorded as technical assistance over the

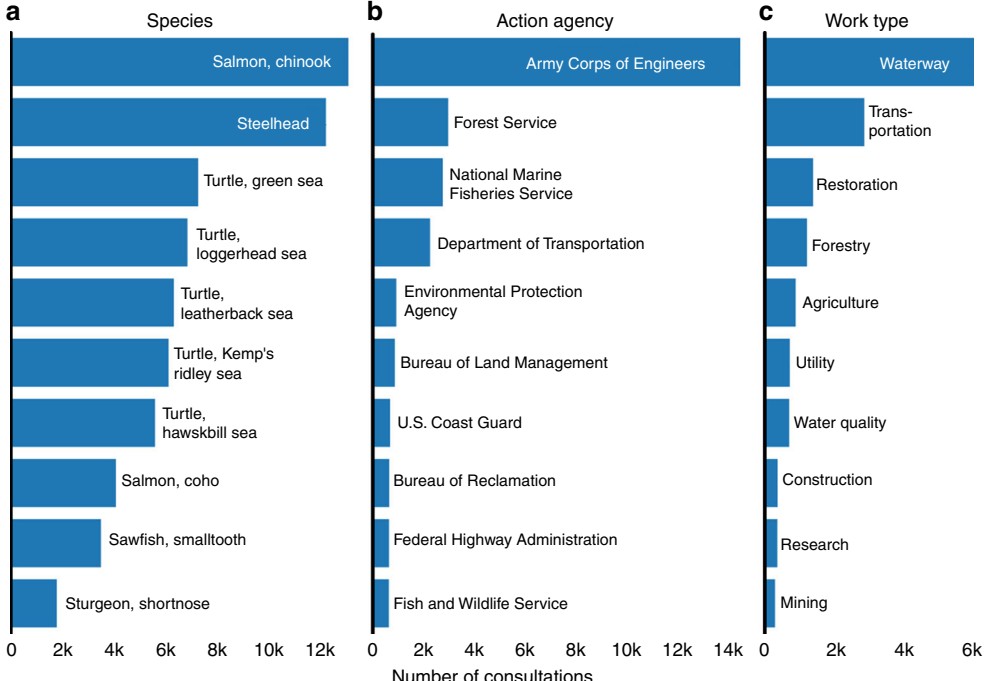

**Fig. 1** Frequencies of U.S. Endangered Species Act section 7 consultations conducted by the National Marine Fisheries Service involving different species (**a**), federal action agencies (**b**), and work types (**c**) between 2000 and 2017. The ten most frequent members of each group are shown. Species listed from top to bottom are *Oncorhynchus tshawytscha, Oncorhynchus mykiss, Chelonia mydas, Caretta caretta, Dermochelys coriacea, Lepidochelys kempii, Eretmochelys imbricata, Oncorhynchus kisutch, Pristis pectinate, Acipenser brevirostrum*

same period. Consultations were unevenly distributed among species ($X^2_9 = 11{,}872.6$, $p < 0.001$), federal agencies ($X^2_9 = 69{,}853.0$, $p = < 0.001$), and work types ($X^2_9 = 19{,}185.7$, $p < 0.001$). The species most commonly consulted on (Fig. 1a) were chinook salmon (*Oncorhynchus tshawytscha*) and steelhead trout (*Oncorhynchus mykiss*). The most frequently consulting agency was the Army Corps of Engineers, with a consultation rate ~6 times higher than the next-closest agency, the Forest Service (Fig. 1b). The most common work type requiring consultation was 'waterway' (Fig. 1c), which includes activities like flood control, bank stabilization, dredging, and dock construction. While the number of informal consultations increased over time ($\Delta per\ year = 28.71$, $SE = 7.21$, $F_{16,1} = 15.84$, $p = 0.001$), the number of formal consultations remained relatively constant ($\Delta per\ year = -0.53$, $SE = 2.46$, $F_{0,1} = 0.05$, $p = 0.833$).

**Agreement between federal agencies and the Services**. To evaluate how often action agencies' determinations aligned with NMFS species experts' determinations, we compared agencies' proposed determinations to the NMFS final determinations. In cases of disagreement, we assume the NMFS analysis to be more accurate because NMFS is the expert wildlife agency, although this assumption may not be true in all cases. Weighted Kappa statistics indicated moderate agreement between NMFS and action agencies ($K_w = 0.38$), and NMFS agreed with the majority (79%) of action agency proposed determinations (Table 1). In cases of discrepancy, action agencies underestimated the effects of proposed actions more frequently (71%) than they overestimated effects (Fig. 2). The most common form of discrepancy occurred when an action agency proposed an NLAA determination and NMFS subsequently made an LAA determination (indicated by a jeopardy or a no jeopardy determination; Table 1).

Federal agencies differed in the degree and type of disagreement with NMFS on the effects of proposed actions (Fig. 2). Among action agencies with at least 20 determinations, the

Federal Emergency Management Agency both over- and under-estimated effects ($K_w = -0.014$, $D_{52} = 0.12$, $p = 0.028$). The Bureau of Land Management ($K_w = 0.73$, $D_{195} = 0.07$, $p = 0.016$), Forest Service ($K_w = 0.64$, $D_{469} = 0.07$, $p < 0.001$), and NMFS ($K_w = 0.65$, $D_{1525} = 0.05$, $p < 0.001$) agreed with NMFS more often than were other agencies on average (Fig. 2). The Environmental Protection Agency ($K_w = -1.00$, $D_{122} = 0.37$, $p < 0.001$) and National Park Service ($K_w = 0.23$, $D_{39} = 0.16$, $p = 0.005$) tended to underestimate the effects of proposed actions. Conversely, the Army Corps of Engineers ($K_w = 0.29$, $D_{2508} = 0.04$, $p < 0.001$) and Federal Energy Regulatory Commission ($K_w = 0.20$, $D_{83} = 0.13$, $p < 0.001$) tended to overestimate effects.

**Consultation outcomes**. Of 4934 formal consultations, 72 (1.5% of formal and 0.3% of all consultations) resulted in jeopardy findings and 55 (1.1% of formal, 0.2% of all consultations) resulted in findings of adverse modification of critical habitat. These consultations consisted of 641 jeopardy and 503 adverse modification determinations. Three consultations resulted in adverse modification without jeopardy and 37 resulted in jeopardy without adverse modification. All projects could proceed if the permittee adopted reasonable and prudent alternatives to minimize or partially offset the adverse effects of the project.

To understand the causes and consequences of the differing determinations that ultimately concluded with jeopardy, we explored jeopardy and adverse modification conclusions in greater detail. Rates of jeopardy determinations differed among species ($X^2_9 = 16.15$, $p = 0.064$), and rates of jeopardy consultations differed among work categories ($X^2_8 = 153.69$, $p < 0.001$). Federal actions related to fisheries management and pest control were more likely to result in jeopardy than other work types (Supplementary Fig. 2a). Among species with at least 10 consultations, the Cook Inlet DPS of beluga whale (*Delphinapterus leucas*) had the highest rates of jeopardy determinations (Supplementary Fig. 2b), although Pacific salmonid species had

**Table 1 Frequencies of determinations proposed by action agencies vs. final determinations made by NMFS during section 7 consultation from 2000 to 2017**

| | Action agency determinations | No effect | NLAA | LAA | No jeopardy (Proposed spp.) | Jeopardy (Proposed spp.) |
|---|---|---|---|---|---|---|
| NMFS Determinations | No Effect | 3671 (**0**) | 7215 (+**1**) | 1377 (+**2**) | 6 (+**2**) | 3 (+**3**) |
| | NLAA | 853 (−**1**) | 59,258 (**0**) | 2509 (+**1**) | 5 (+**1**) | 0 (+**2**) |
| | No Jeopardy (LAA*) | 163 (−**2**) | 2246 (−**1**) | 13,454 (**0**) | 17 (**0**) | 0 (+**1**) |
| | Jeopardy (LAA*) | 55 (−**3**) | 164 (−**2**) | 439 (**0**) | 0 (−**1**) | 0 (**0**) |

Bold numbers show the 'discrepancy' score assigned to a given combination, indicating the degree of agreement (positive values) or disagreement (negative values)
*NMFS must make a Jeopardy/No jeopardy determination following an 'LAA' finding during informal consultation

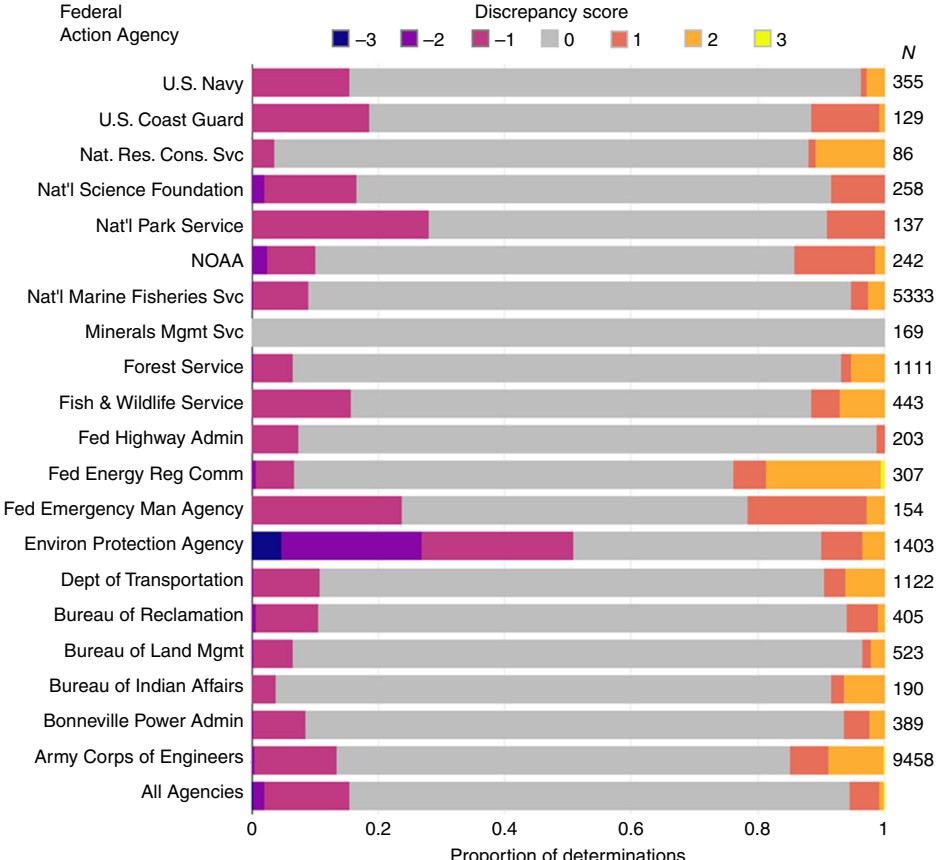

**Fig. 2** Ordinal 'discrepancy' scores indicating the degree of disagreement on determinations between U.S. federal agencies and the National Marine Fisheries Service (NMFS) during section 7 consultations between 2000 and 2017. Positive values indicate an overestimation of effects by an agency, and negative values indicate underestimation. Bar length represents the percentage of determinations by an agency receiving each score, and 'N' provides the number of determinations. Federal agencies varied in rates of disagreement with NMFS during section 7 consultation under the U.S. Endangered Species Act. NMFS and the Fish and Wildlife Service may disagree with themselves because within each agency different divisions are responsible for implementing management actions and conducting section 7 consultations on those actions

the greatest number of jeopardy determinations (Table 2). The rate of consultations that ended in jeopardy was constant over time ($\Delta per\ year = -0.001$, $SE = 0.001$, $F_{1,15} = 4$, $p = 0.640$).

To evaluate whether particular types of actions disproportionately led to jeopardy determinations, we tested for over- or under-representation of species-work type combinations among jeopardy determinations. There was a disproportionately high number of jeopardy determinations resulting from proposed actions categorized as "agriculture" affecting chinook salmon ($Effect = 8.6$, $sd = 2.8$, $p = 0.003$), coho salmon ($Effect = 10.9$, $sd = 2.4$, $p < 0.001$), and steelhead ($Effect = 8.7$, $sd = 2.7$, $p < 0.001$; Fig. 3). The agriculture work type includes pesticide

registration, irrigation, and grazing allotment decisions. Blue (*Balaenoptera musculus*), humpback (*Megaptera novaeangliae*), fin (*B. physalus*), North Atlantic right (*Eubalaena glacialis*), sei (*B. borealis*), and sperm whales (*Physeter macrocephalus*), as well as leatherback sea turtles (*Dermochelys coriacea*), were disproportionately jeopardized by actions in the "fishery" category ($Effect > 1.5$, $sd = 0.6–0.9$, $p < 0.05$; Fig. 3). Finally, ringed seal (*Phoca hispida*) were disproportionately jeopardized by actions categorized as "utility" ($Effect = 1.0$, $sd = 0.2$, $p = 0.038$; Fig. 3), which includes hydropower, pipeline, and transmission line construction and maintenance, and gulf sturgeon (*A. oxyrinchus desotoi*) by actions categorized as "ocean" ($Effect = 1.0$, $sd = 0.2$,

| Table 2 The ten species with the greatest number of jeopardy determinations in formal NMFS section 7 consultations between 2000 and 2017 | | | |
|---|---|---|---|
| Common name | Jeopardy determinations | All determinations | % |
| Green sea turtle (*1/5 DPS*) | 9 | 778 | 1.2 |
| Kemp's ridley sea turtle | 9 | 614 | 1.5 |
| Killer whale (*1/1 DPS*) | 9 | 186 | 4.8 |
| Loggerhead sea turtle (*2/8 DPS*) | 10 | 768 | 1.3 |
| North Atlantic right whale | 10 | 176 | 5.7 |
| Leatherback sea turtle | 11 | 703 | 1.6 |
| Chum salmon (*2/4 ESU*) | 16 | 564 | 2.8 |
| Sockeye salmon (*2/7 ESU*) | 24 | 675 | 3.6 |
| Coho salmon (*5/7 ESU*) | 31 | 1490 | 2.1 |
| Steelhead (*13/15 DPS*) | 60 | 3012 | 2.0 |
| The number of distinct DPS/ESUs involved in jeopardy determinations is shown parenthetically | | | |

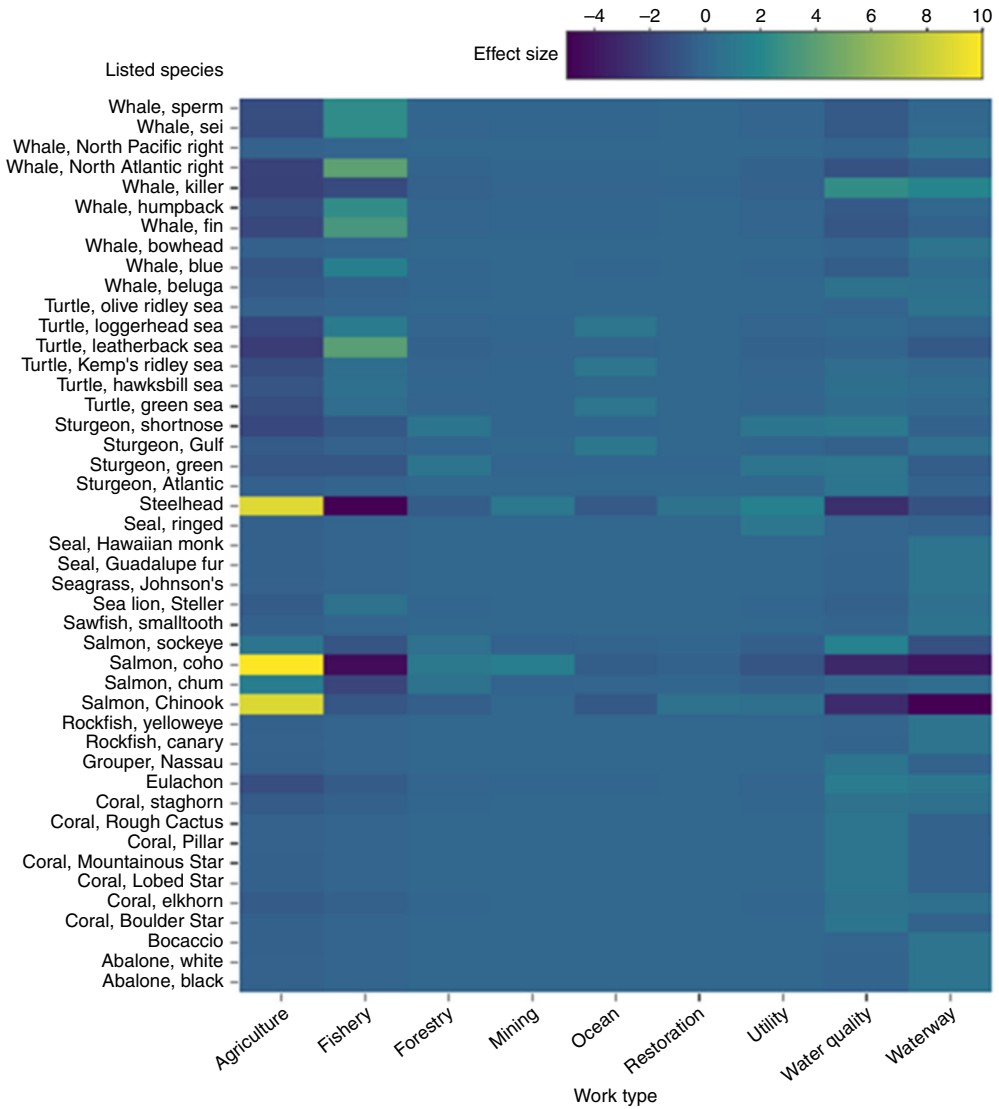

**Fig. 3** Heatmap displaying differences between the observed and expected frequency of jeopardy determinations for combinations of species and work types during U.S. Endangered Species Act section 7 consultation between U.S federal agencies and the National Marine Fisheries Service from 2000 to 2017. Higher values indicate combinations for which jeopardy determinations occurred more frequently than would be expected under a random association

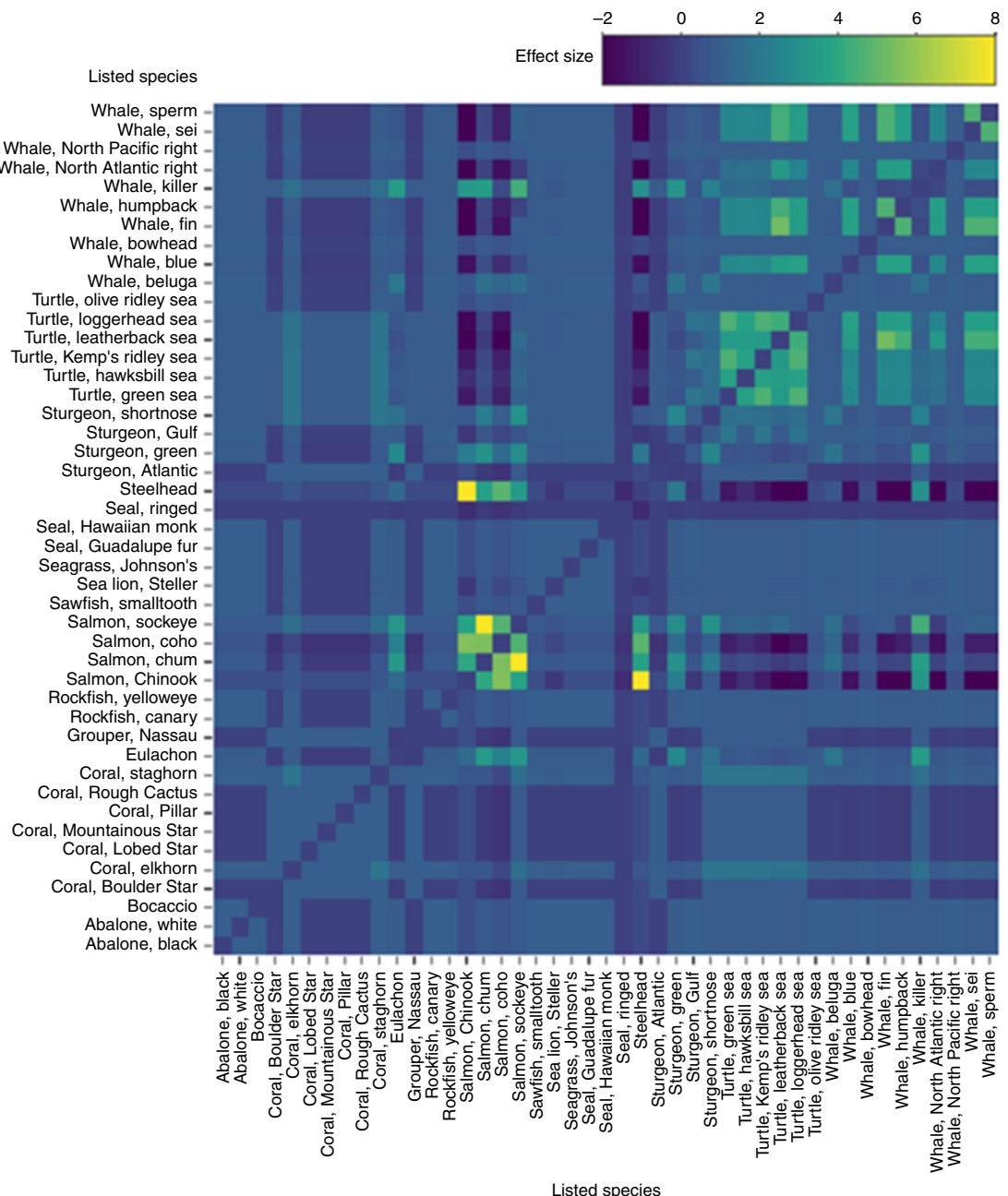

**Fig. 4** Heatmap displaying differences between the observed and expected frequency with which jeopardy determinations were made for pairs of species during the same U.S. Endangered Species Act section 7 consultation between U.S federal agencies and the National Marine Fisheries Service from 2000 to 2017. Higher values indicate species pairs for which observed co-jeopardizations were greater than would be expected under a random association

$p = 0.030$), which includes shoreline stabilization, geotechnical exploration, and waste disposal. Both coho salmon and steelhead were jeopardized less than expected by fishery actions (*Effect* < $-4.5$, *sd* = $1.9–2.2$, $p < 0.007$), and both chinook and coho salmon less than expected by waterway actions (*Effect* < $-4.0$, *sd* = $2.1–2.6$, $p < 0.05$; Fig. 3).

Chinook salmon and steelhead had the highest probability (0.42) of being jeopardized by the same action (co-jeopardization) among all species (Fig. 4), and all Pacific salmonids exhibited significant co-jeopardization (*Effect* > 2, $p < 0.024$). Additionally, the Southern Resident DPS of killer whale (*Orcinus orca*) were co-jeopardized with all Pacific salmonids (*Effect* > 4.4, $p < 0.018$), except coho salmon. Finally, significant co-jeopardization occurred between green sturgeon (*Acipenser medirostris*) and sockeye salmon (*Effect* = 2.9, $p = 0.009$), and eulachon

(*Thaleichthys pacificus*) and coho salmon (*Effect* = 2.4, $p = 0.024$; Fig. 4).

## Discussion
Using data to critically evaluate the efficacy and efficiency of laws and regulations can help clarify contentious topics and guide the development of future policy. The U.S. Congress recently passed the Foundations for Evidence-Based Policymaking Act specifically to help ensure that federal decisions are based on data rather than conjecture[17]. Our analysis of NMFS consultation data provides the first quantitative evidence of the importance of having species experts evaluate the potential effects of proposed federal actions, rather than relying solely on action agency staff. The data illustrate that even though the clear majority (99.7%) of federal

actions proceed without substantial changes because of a jeopardy finding, rare instances of jeopardy determinations made by species experts at NMFS were critical to ensuring that federal agencies did not authorize actions that would jeopardize listed species. Consultation between federal agencies and species experts at the Services is one of the most important provisions in the ESA, and often the most controversial[18,19]. Results indicate that recommendations to reduce the role of expert agencies in the consultation process may compromise the conservation of imperiled species and point to possible approaches to improve the efficiency of consultation without sacrificing species protections.

Our results show that excluding the species experts at NMFS from the consultation process could have been detrimental to the conservation of certain threatened and endangered species. The purpose of the ESA is to conserve imperiled species, and section 7 consultations are the primary mechanism through which the ESA ensures that federal agencies do not compromise this purpose[20]. While NMFS agreed with most action agency proposed determinations, agreement rates varied substantially depending on the action agency, type of action, and species. The potential impact of erroneous consultation outcomes is severe: if action agencies had been allowed to self-consult, actions resulting in almost one-third (219) of the jeopardy determinations made by NMFS would not have received thorough biological analysis because the action agencies had made a no effect or NLAA determination. Without NMFS involvement, these instances would have authorized 22 actions that jeopardized 14 species, many of which are economically important (e.g., commercially harvested salmonids). Thus, without the evaluation provided by an expert agency, there may have been numerous instances of federal agencies violating (albeit perhaps unintentionally) their duty to ensure their proposed actions would not jeopardize a listed species or destroy or adversely modify critical habitat.

The details of cases of disagreement between action agencies and NMFS offer insights for future conservation policy. These cases were primarily limited to a subset of species and work types in which wide-ranging actions had the potential to adversely affect multiple, spatially-overlapping species. The federal registration of pesticides (categorized under the agriculture work type) made up a disproportionate percentage of jeopardy findings for multiple species of Pacific salmonids, and the authorization of Atlantic fisheries plans made up a disproportionate percentage of jeopardy findings for several whale and turtle species. The spatial extent of these proposed actions and the spatial overlap of affected species meant that such actions would have potentially severe consequences if their effects were underestimated. This emphasizes the importance of involving species experts in the consultation process. Most federal agencies do not have the same biological expertise or dedicated resources to conduct an equally thorough and informed evaluation of the conservation impacts of their actions as do agencies with dedicated imperiled species biologists, such as NMFS. Furthermore, it is likely that agencies whose priority is not the protection of imperiled species may be motivated to expedite projects that fulfill their institutional mission. Therefore, checks for potentially harmful federal actions—like those currently provided by expert evaluation—are crucial for conservation laws like the ESA to prevent the extinction of species.

Patterns of concurrence between action agencies and NMFS can also be used to identify and inform opportunities for increased efficiencies in the section 7 consultation process that do not sacrifice species protections. Because an LAA finding triggers the expenditure of additional effort for formal consultation, providing clear guidance at this stage might reduce the consultation workload considerably without undermining conservation. The LAA determination was the most common type of discrepancy between NMFS and action agencies, accounting for

53% of disagreements, and presents an opportunity to improve consultation efficiency. The best example of this was the case of waterway activities. These actions were most commonly initiated by the Army Corps of Engineers and Federal Emergency Management Agency, which often overestimated the effects of proposed actions. Such overestimation resulted in 159 unnecessary formal consultations because NMFS ultimately concluded that the species were not affected or were unlikely to be adversely affected. This is an example where explicit standards for LAA thresholds could reduce instances of disagreement and conserve agency resources. Policy guidance to address the current lack of detailed, quantitative standards for the LAA threshold could be very fruitful. Interim approaches for pesticide assessment[21] and the consultation keys for woodstork[22] under the ESA provide two different examples of how the Services have clarified the LAA threshold, and potential models for the development of conservation laws implementing an efficient consultation system.

Patterns of jeopardy determinations among species-work type combinations provide evidence for the benefit of advanced planning to improve both species outcomes and consultation efficiency. The best example comes from actions related to fisheries management, which resulted in fewer jeopardy determinations than expected for multiple Pacific salmonid species. On the surface, this result was surprising, because we expect these types of actions to negatively affect anadromous species. However, in the case of fishery management, a rule issued under section 4(d) of the ESA allows commercial use of some listed salmonid species[23] and sets quantitative standards for developing and approving fishery management plans for these species. This process front-loaded much of the analysis of effects for similar actions, expediting subsequent consultation and reducing the probability that proposed actions would jeopardize the species. Conducting advanced, programmatic consultation likely provide opportunities for efficient and effective implementation of conservation laws.

The NMFS data show that jeopardy and adverse modification determinations are very rare, and we know of no instance in which such a determination stopped a project because alternatives were unavailable. We note, however, that the very low rates of jeopardy and adverse modification from NMFS (<2% of formal consultations) are higher than those from FWS (<0.1% of formal consultations)[12]. One possible explanation is that DPS/ESUs are more common among species managed by NMFS than FWS, and the effects of proposed actions may be more likely to cross a jeopardy threshold for these smaller listed units than for subspecies or full species. Notably, all Pacific salmonids, which were the majority of species involved in jeopardy determinations, are divided into multiple DPS/ESUs. NMFS also manages fewer species and conducts fewer consultations than FWS (24,893 from 2000 to 2017 vs 88,290 from 2008 to 2015). This may provide the agency with greater bandwidth to analyze project effects and thus increase its confidence in finding and defending jeopardy determinations. Finally, differences between NMFS's and FWS's history and approach to consultations may explain some of our results[24]. Future research should evaluate the degree to which these factors are responsible for differences in how the ESA is implemented between the Services. While the causes of differences between the Services may be unclear, the low percentage of jeopardy and adverse modification findings shows that NMFS, like FWS, has worked with agencies and applicants to find solutions the vast majority of the time. Our results underscore the same message as research using parallel consultation data from the FWS: conventional wisdom about the ESA stopping projects is unfounded[11].

To achieve better outcomes for more species, conservationists have explored for decades more efficient approaches to administering conservation laws. Administrative data are a key yet under-used resource for understanding the strengths and weaknesses of laws and policies and can be used to make their implementation more effective. We found that minimizing the involvement of expert agencies for the ESA could threaten the very existence of many listed species. This finding provides a stark illustration of the pitfalls of making policy decisions without data. Data describing the implementation of conservation programs, like those from NMFS evaluated here, provide critical insight into the reality of implementation and can inform regulatory policies (e.g., quantitative LAA guidelines) that may improve conservation outcomes for imperiled species and make conservation laws more efficient and less contentious to implement. We were unable to systematically assess the effectiveness of consultations, or whether NMFS determinations were more correct than those of action agencies, because species status and/or consultation outcome data were lacking. Such data would enable an objective assessment of consultation conclusion accuracy in addition to the comparisons between evaluators presented here. This short-coming underscores the fact that data must be recorded and curated before they can be analyzed and used. As the number of endangered species continues to grow, data-driven strategies become increasingly urgent if funding for environmental protection continues to stagnate[25,26]. Especially in the face of funding shortcomings, the use of data to guide policy and administrative decisions, rather than conjecture and anecdotes, can create effective and efficient approaches to conserve biodiversity.

## Methods

**Data preparation.** We obtained data from all formal and informal consultations as recorded in the PCTS database by NMFS biologists through June 2017. In addition to the species involved in a consultation and the determinations made by NMFS, PCTS records include the action agency, category of proposed action, dates of consultation initiation and conclusion, and the determinations proposed by action agencies. See supplemental Table S1 for a full list and description of fields.

Because records prior to 2000 were deemed potentially unreliable based on the frequency of data recorded and conversations with NMFS personnel, we analyzed data from 2000 to 2018. We performed several quality control steps to correct errors that may have accumulated from >2000 agency staff entering data over several decades. We corrected apparent date errors (e.g., end dates earlier than start dates) and homogenized the names of species, action agencies, and work types. NMFS records a variety of information about the nature of consultations in a single Consultation Type field. We split this into a Type field that indicates whether a consultation was recorded as formal, informal, or combined and a Complexity field that indicated whether a consultation was standard, programmatic, conference, or early.

Species and critical habitat determinations are recorded in a variety of combinations in PCTS. We standardized these outcomes by re-coding species determinations into one of four categories: 'no effect', 'NLAA', 'no jeopardy', or 'jeopardy'. We re-coded critical habitat determinations into 'no effect', 'NLAA', 'no adverse modification', or 'adverse modification'. We coded determinations for species that did not have critical habitat designated at the time of the consultation as 'no critical habitat'. To ensure that all reported instances of jeopardy or adverse modification were accurate, we examined the biological opinions for these consultations and recorded proposed and final determinations, as well as work categories. Thus, our results reflect the minimum number of jeopardy determinations as there may have been erroneous non-jeopardy determinations recorded. In addition, we manually inspected 320 consultations for which outcomes were unclear based on PCTS records. Although this large dataset likely contains additional minor errors that we were unable to correct, we assume that those errors are unbiased and randomly distributed within the data.

**Data analyses.** All statistical analyses were performed in R 3.5.1[27]. We estimated changes in consultation frequency over time by fitting linear models with a log link and Poisson error distribution to the number of consultations recorded as informal and formal as a function of year. NMFS is organized into five geographic regions: Northeast, Southeast, Alaska, Pacific Island, and West Coast. We used a Chi-square test to estimate differences in formal consultation rates among geographic regions. Prior to 2013, the West Coast region consisted of the Southwest and Northwest regions, and we aggregated all consultations from these regions into a single West Coast category for consistency across years.

We also tested for differences in the frequency of consultation among species, action agencies, and work type using Chi-square goodness-of-fit tests, including only consultations for which a species was recorded. Out of 116 species consulted on by NMFS, 59 species had distinct population segments (DPSs) or evolutionarily significant units (ESUs) designated. For our analysis of species-specific consultation frequencies, we considered all DPS/ESUs of a given species together. For instance, a consultation involving multiple coho salmon (*Oncorhynchus kisutch*) DPSs was counted as a single consultation for coho salmon.

To evaluate patterns of agreement between NMFS and action agencies, we tabulated the frequencies of all possible combinations of determinations proposed by action agencies and those made by NMFS. We used weighted Kappa statistics ($K_w$) to indicate the overall degree of agreement between NMFS and federal agencies. We modified the weight matrix to reflect the differing magnitudes of each type of disagreement. Possible weights included the set {0, 0.25, 0.50, 0.75, 1}, with cells representing agreement receiving a weight of 0. These included situations in which NMFS and the action agency both determined no effect or NLAA, or when an action agency determined LAA and NMFS subsequently made either a jeopardy or no jeopardy determination. To identify agencies exhibiting extreme rates of over- or underestimation of effects, we also created an ordinal 'discrepancy' variable to rank the degree of disagreement between action agencies and NMFS. Determinations for which the action agency underestimated effects were assigned a negative score, while those in which the action agency overestimated effects were assigned a positive score (Table 1). Instances of agreement, as defined above, were assigned a score of '0' (Table 1). We then used a two-tailed Kolmogorov–Smirnov tests to compare the distribution of discrepancy scores for a given agency against the distribution of discrepancy scores among all agencies. An agency exhibited significant departure from overall rates of disagreement if the probability of the test statistic $D$ was $p < 0.05$. We restricted these analyses of discrepancy to agencies with at least 20 recorded consultations.

We estimated the rate of jeopardy determination made during formal consultations and the proportion of formal consultations with at least one jeopardy determination and tested for changes over time using generalized linear models with an identity link and normal error distribution. We tested for differences in jeopardy determination rates among species and for the proportion of consultations with a jeopardy conclusion among work types using Chi-square goodness-of-fit tests. We used matrix permutation to identify combinations of species and work categories that exhibited a disproportionately high frequency of jeopardy determinations. First, we constructed a matrix containing the frequency of jeopardy determinations for every combination of species and work type. To create a null distribution for these frequencies, we then randomized cell counts 1000 times while keeping row and column totals fixed using the *vegan* package[28] in R. The probability that an observed frequency was greater than random chance was calculated as the proportion of permutations in which the simulated frequency was greater than the observed frequency. We considered combinations with $p < 0.05$ to exhibit significant positive association. We report effect sizes as the difference between the observed and mean simulated cell frequencies, and the standard deviation of simulated cell frequencies.

Finally, a consultation determining jeopardy for one species may be more likely to also reach a jeopardy determination for other closely related and/or spatially proximate species. We quantified rates at which pairs of species were jeopardized by the same proposed action (i.e., in the same consultation), which we referred to as co-jeopardization. Rates of co-jeopardization that were greater or less than random were determined using a matrix permutation test. We organized consultation data into a binary species by consultation matrix in which cells indicated whether a species was jeopardized in a consultation. We estimated the pairwise probabilities of co-jeopardization and effect sizes (i.e., the difference between observed and expected frequency of co-jeopardization) for species with at least one jeopardy determination using the *cooccur* package[29] for R. *Cooccur* does not produce measures of uncertainty for effect size estimates. We considered pairs of species for which the proportion of permutations resulting in co-jeopardization was greater than observed $p < 0.05$ to exhibit significant association.

**Reporting summary.** Further information on research design is available in the Nature Research Reporting Summary linked to this article.

## Data availability

The consultation data that support the findings of this study, and all R code used to conduct statistical analyses and create graphs are available in a public Open Science Framework repository (10.17605/OSF.IO/UCFKJ).

## Code availability

All code used to conduct analyses in this study is publicly available through an Open Science Framework repository (10.17605/OSF.IO/UCFKJ)

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

## Acknowledgements

We thank C. Tortorici and K. Petersen for providing access to the PCTS database. J. Miller, J. Rappaport-Clark, and R. Dreher provided review and feedback on drafts of the manuscript.

## Author contributions

M. Evans conducted data analysis and lead manuscript writing. J. Malcom and Y-W. Li conceptualized the study, assisted with study design, and contributed to manuscript writing.

## Additional information

**Competing interests:** The authors declare no competing interests.

