## [Peer Review File · Nature Communications]

Reviewer #1 (Remarks to the Author):

The issue of how species at risk and their habitats are protected is of international interest. This paper focuses on an ongoing debate in the United States about the consultation process for the subset of species and issues that fall under the consultative jurisdiction of the National Marine Fisheries Service (NMFS). Through examination of the administrative record, the authors show that a small percentage of cases where actions proposed by “action agencies” (e.g., US Army Corps of Engineers, US Forest Service) result in determinations at odds with the conclusions of expert consultation provided by NMFS.

This finding is of interest to those assessing the validity and effectiveness of the administrative process, which the authors conclude is working because differences occur in a small percentage of cases, where NMFS may catch proposed actions that would further endanger species listed under the ESA.

The authors suggest that these findings support the practice of consultation by expert agencies, such as NMFS. They note that the consultation process (under Section 7 of the ESA), is under attack by those wishing to streamline the review of proposed actions that might have detrimental effects on listed species. While this is an important point, the manuscript does not address the effectiveness of the overall consultative process - is it good, or just better than a weakened version of the current system. For example, if consultation by expert agencies is helpful, would more consultation be even more helpful? Are there other organizations or scientific bodies that might be better or more efficient? Is the legal process often invoked by litigation of controversial cases an additional safeguard or a hindrance to effective process?

I found the concluding section of the manuscript to be interesting, but the authors' arguments stray from the results of their analysis. While the manuscript provides “critical insight into the reality of implementation and can inform regulatory policies,” as the authors state, it does not offer data to support their suggestion that the management agencies make determinations “without data.” This may be true in some cases, perhaps far too many. But the unsupported implication overshadows the results of the analysis. I was left wondering whether the same databases that the authors consulted might also support an assessment of the quality or correctness of the action agencies' determinations, overall, not just the cases where they disagreed with the expert agencies. Without some sort of assessment of error, based on the needs of the threatened and endangered taxa, the authors' call for “the use of data rather than conjecture and anecdotes to guide policy and administrative decisions” seems off-point.

I suggest that the authors consider incorporating the findings from this manuscript into an expanded treatment of the ESA consultation process. Readers that understand the implications for the administration of the ESA may be more inclined to base their actions on a comprehensive treatment, which would also ground the political and policy debate centering on Section 7 consultations.

Reviewer #2 (Remarks to the Author):

Review of 'Novel data show expert wildlife agencies are indispensable to protecting endangered species'

The authors use a database from the US National Marine Fisheries Service (NMFS) to investigate levels of agreement of risk to species caused by US federal government actions as classified under the Endangered Species Act (ESA). They specifically look at how US federal agencies and NMFS evaluate risk to species under Section 7 of the ESA. They find that in general there is good agreement between federal agencies and NMFS, though in the cases of differences, federal agencies tend to under-estimate risk. This is an interesting and important evaluation, and provides a useful conclusions to enable evidence-based decision making around ESA implementation.

That being said, I do have some suggested improvements for the study. I feel that the overall framing and discussion seems to emphasize the times when there is disagreement between federal agencies and the NMFS. Yet, the results seem to show that there is actually high degree of agreement. I would suggest kappa statistics or similar could be used to more clearly quantify the levels of agreement, as this is the norm used for these sorts of comparisons.

The results also seem very wordy, considering many different aspects of what consultations were conducted. I would suggest more succinctly summarizing the levels of agreement and then more quickly and strongly getting into the details of where there are differences between NMFS assessments and the federal agency assessments, and looking at what decisions the agencies are getting wrong. Is it possible to predict which types or species or decisions are likely to be the most problematic? I know the authors do address this, but it feels very buried in the results.

The authors also need to be clearer about the difference between a 'no effect' finding and NLAA. This isn't introduced in Box 1. Table 1 also feels slightly confused to me, as there's no 'LAA' listed for NMFS. I wonder whether separating this into two tables to consider the 'no effect', 'LAA', and 'NLLA' as one table, and then what happens with the jeopardy findings separately. As for a jeopardy finding to be there, my understanding is that NMFS would have had to report LAA from the informal consultation? Therefore I think the jeopardy data in table 1 is really a subset of the LAA/NLAA/no effect data.

Firstly, there is an assumption that the NMFS are always correct. I wonder whether there has been any evaluation of accuracy of NMFS, even at a smaller scale? To be able to cite this would strengthen the case for this study. If not, is there any way to take a smaller sample of decisions you identified where there's disagreement between federal agencies and NMFS, and then have an independent third party evaluate (without knowing what the NMFS and federal agency decided) to see if they reach the same conclusions?

Finally, a couple of language issues. This manuscript is written with some very emotive language in the abstract and even in some parts of the main manuscript e.g. 'The strongest legislation any nation has enacted to conserve imperiled species...' - this needs a reference that justifies this, or toning down slightly. I would suggest more cautious language in some places. It's also written from a very US-centric view. With references to 'Congress' etc, and just jumping into descriptions of the NMFS etc without any reference to it being part of the US government, and that the ESA applies only in the US. While clearly the policy relevance of this piece is in the US, I think the framing needs to be adjusted given this is submitted to an international journal for an international audience.

NB - as a reviewer I would much appreciate it if you could include page numbers and line numbers in manuscripts for review, as it makes it easier to refer to specific text sections.

Response to Reviewer Comments on Manuscript # NCOMMS-18-32702

Reviewer 1

I found the concluding section of the manuscript to be interesting, but the authors arguments stray from the results of their analysis. While the manuscript provides “critical insight into the reality of implementation and can inform regulatory policies,” as the authors state, it does not offer data to support their suggestion that the management agencies make determinations “without data.” This may be true in some cases, perhaps far too many. But the unsupported implication overshadows the results of the analysis.

We do not mean to suggest that any agency, expert or otherwise, make consultation determinations without data. Rather, data that can inform national conservation policies has been lacking. We have edited several lines in this paragraph to clarify this point as follows: “Especially in the face of funding shortcomings, the use of data to guide policy and administrative decisions, rather than conjecture and anecdotes, can create effective and efficient approaches to conserve biodiversity.”

I was left wondering whether the same databases that the authors consulted might also support an assessment of the quality or correctness of the action agencies’ determinations, overall, not just the cases where they disagreed with the expert agencies. Without some sort of assessment of error, based on the needs of the threatened and endangered taxa, the authors’ call for “the use of data rather than conjecture and anecdotes to guide policy and administrative decisions” seems off-point.

Without regular and comprehensive species status data it is difficult, if not impossible to make this kind of assessment. Unfortunately, this data is lacking from the section 7 database, and does not exist in any other centralized repository to our knowledge. We have added a line in the concluding section acknowledging this shortcoming and use it to illustrate the importance of generating data in the first place: “We were unable to systematically assess the quality or effectiveness of consultations overall, due to the lack of necessary species status and/or consultation outcome data. This highlights the importance of thorough data recording and curation even before it can be used.”

We additionally inserted a clause in the introduction acknowledging this limitation as well: “While we could not evaluate the outcome of consultations, these questions are critical for understanding the conservation impacts of the consultation program and predicting the effects of proposed changes.”

I suggest that the authors consider incorporating the findings from this manuscript into an expanded treatment of the ESA consultation process. Readers that understand the implications for the administration of the ESA may be more inclined to base their actions on a comprehensive treatment, which would also ground the political and policy debate centering on Section 7 consultations.

We received feedback from Reviewer #2 to broaden the context of the manuscript, and scope of the discussion, beyond ESA specific implications. This is at odds with a more comprehensive treatment of the ESA consultation process while maintaining a concise and focused discussion. Given Nature Communications' international and interdisciplinary readership, we have prioritized a broader scope.

Reviewer 2

I feel that the overall framing and discussion seems to emphasize the times when there is disagreement between federal agencies and the NMFS. Yet, the results seem to show that there is actually high degree of agreement.

We have changed the order of our reporting of instances of agreement and disagreement in the Results section to emphasize the degree of agreement, and re-written the summary of these findings in the Abstract to reflect the frequency of this outcome as follows: "...federal agencies and NMFS frequently agreed (79%) on how federal actions would affect listed species."

Similarly, we have rearranged the structure of several Discussion paragraphs to better portray agreement as the norm and instances of disagreement as important exceptions.

I would suggest kappa statistics or similar could be used to more clearly quantify the levels of agreement, as this is the norm used for these sorts of comparisons.

We have calculated and now report weighted Kappa statistics indicating the degree of agreement between NMFS and action agencies, both in aggregate and individually. We retain the use of Kolmogorov-Smirnoff statistics as complimentary indicators of levels of agreement, because this approach provides a probabilistic threshold for identifying departure from agreement.

The results also seem very wordy, considering many different aspects of what consultations were conducted. I would suggest more succinctly summarizing the levels of agreement and then more quickly and strongly getting into the details of where there are differences between NMFS assessments and the federal agency assessments, and looking at what decisions the agencies are getting wrong. Is it possible to predict which types or species or decisions are likely to be the most problematic? I know the authors do address this, but it feels very buried in the results.

We have re-organized paragraphs in the Results section to bring more focus on the patterns of agreement and instances of discrepancy between action agencies and NMFS. These results are now presented in the second paragraph, which we divide into two sections to make the agency-specific results more accessible. The preceding paragraph has been truncated and similarly restructured to provide only basic results about the most common species, work types and agencies involved in consultation.

The authors also need to be clearer about the difference between a 'no effect' finding and NLAA. This isn't introduced in Box 1.

We have created an additional figure outlining the consultation process, and the potential end points from both the perspective of the Services and action agencies. This makes clear the idea that ‘no effect’ and ‘NLAA’ are two different outcomes.

Table 1 also feels slightly confused to me, as there’s no ‘LAA’ listed for NMFS. I wonder whether separating this into two tables to consider the ‘no effect’, ‘LAA’, and ‘NLAA’ as one table, and then what happens with the jeopardy findings separately. As for a jeopardy finding to be there, my understanding is that NMFS would have had to report LAA from the informal consultation? Therefore I think the jeopardy data in table 1 is really a subset of the LAA/NLAA/no effect data.

Informal consultation data are unavailable, preventing a clean delineation of the consultation process and comparison of conclusions at each stage. However, we have made several changes to Table 1 to clarify the set of possible consultation outcomes and how they relate to intermediate determinations. The fact that ‘No Jeopardy’ and ‘Jeopardy’ determinations are the result of an intermediate ‘LAA’ finding by NMFS is indicated parenthetically in the row titles, and we also include a brief description of this process in a table footnote.

The added supplemental figure also helps to illustrate the possible endpoints of consultation, and demonstrates the interplay between ‘LAA’ findings and ‘Jeopardy/No Jeopardy’ conclusions.

Firstly, there is an assumption that the NMFS are always correct. I wonder whether there has been any evaluation of accuracy of NMFS, even at a smaller scale? To be able to cite this would strengthen the case for this study. If not, is there any way to take a smaller sample of decisions you identified where there’s disagreement between federal agencies and NMFS, and then have an independent third party evaluate (without knowing what the NMFS and federal agency decided) to see if they reach the same conclusions?

We make the assumption that NMFS is more likely to be correct because of the agencies unique expertise with imperiled species’ biology. We have added language to make this assumption explicit, and explain the reasons for it in the Introduction: “Because the Services alone develop plans to recovery listed species and periodically assesses their conservation status, these expert agencies have the greatest understanding of how actions affect the species’ conservation prospects. During consultations, this unique expertise is vital to accurately identifying the effects of an action on species.”

And in the Results: “In cases of disagreement, we assume the NMFS analysis to be more accurate because NMFS is the expert wildlife agency , although this assumption may not be true in all cases.”

To our knowledge, no dataset or evaluation of NMFS accuracy relative to action agencies exists. The suggested analysis using an independent third party to evaluate disparate conclusions made by NMFS and action agencies would be an important and

valuable study in and of itself, but a thorough exploration of this issue is beyond the time and scope of this manuscript.

This manuscript is written with some very emotive language in the abstract and even in some parts of the main manuscript e.g. 'The strongest legislation any nation has enacted to conserve imperiled species...' - this needs a reference that justifies this, or toning down slightly. I would suggest more cautious language in some places.

We have eliminated all instances of emotive language. In particular any reference to the ESA's strength has been toned down. The sentence in the Abstract has been removed, and we have changed the mentioned sentence in the Introduction to read "The Endangered Species Act is among the strongest legislation any nation has enacted to conserve imperiled species."

It's also written from a very US-centric view. With references to 'Congress' etc, and just jumping into descriptions of the NMFS etc without any reference to it being part of the US government, and that the ESA applies only in the US. While clearly the policy relevance of this piece is in the US, I think the framing needs to be adjusted given this is submitted to an international journal for an international audience.

We have clarified where necessary the laws and agencies that are part of the US federal government. More broadly, changes have been made to the Introduction and Discussion sections that reframe the study in a more international scope, using the Endangered Species Act as one example of approaches to conservation policy. For example the last sentence of the opening Introduction paragraph now reads: "Conflicting views raise an important question: how can conservation laws be most effective and cost-efficient to implement?"

Additionally, we begin the following paragraph as follows: "For decades, government agencies, politicians, and the public have offered competing approaches to balancing economic interests with species conservation. One important but controversial approach is to empower dedicated agencies with imperiled species' expertise to evaluate and limit proposed development."

Finally, we have added several references to studies examining international conservation policy and funding in order to connect our results to this broader literature. For example, in the final paragraph of the Discussion we now state: "Despite their strength on paper, the ESA and other conservation laws remain severely underfunded in practice." referring to a Science paper analyzing global conservation funding shortages (McCarthy et al. 2012).

Reviewer #1 (Remarks to the Author):

The manuscript provides a good commentary on some aspects of the consultation process mandated by the United States Endangered Species Act. The authors introduce the need to examine this topic by referencing proposals to streamline ESA procedures by eliminating consultation by government experts outside the agencies regulating actions that might affect listed species adversely. Thus, their MS is largely in response to political calls to weaken this aspect of ESA implementation.

The authors have addressed many issues raised in previous reviews. They have clarified multiple sections and reduced the emotive language prominent in the first submission. The overarching points regarding the limited nature of the data used to evaluate the consultation process cannot be overcome, and the authors have done a good job extracting what they can from a database that does not contain much in the way of scientific details or biological outcomes of the consultation process.

I remain attentive to the fact that this paper focuses on the minority of cases where administratively defined experts disagreed with the agencies overseeing implementation of the ESA. The paper's findings are pertinent to the administrative aspects of policy implementation, but no assessment of the quality or correctness of the agencies' ultimate determinations or actions is possible at this time.

It remains unclear whether the cases of agreement, as well as those of disagreement, advance or inhibit the task of species recovery. In their response to earlier reviews, the authors note that, due to the lack of data, "it is difficult, if not impossible to make this kind of assessment." Nevertheless, this is the content that many readers will be expecting, given that the title purports to show that "expert wildlife agencies are indispensable to protecting endangered species."

I continue to feel that this paper is of interest to readers that already understand its implications for the administration of the ESA; others may find it very narrowly focused on a subset of administrative procedures mandated by the Act. If the authors wish to reach a broader, larger audience, I suggest that they be more direct in describing the contentious political environment in the US, and how current political intervention is putting the Act at risk. This would resonate with international readers, who will then better appreciate the role of expert consultation in the many countries who have modeled their own laws after the US ESA.

Reviewer #2 (Remarks to the Author):

Thank you for responding to my previous review of this manuscript. I recognize that you have addressed many of my previous comments and the manuscript has improved as a result of this. However, I still have some concerns, while many of these are minor, there is still one major concern. Specifically, around the assumption that NMFS determinations are 'correct'. I appreciate that you are unable to verify this with a third party validation, but I would strengthen the text still further about the staff expertise within NMFS, and then reflect on this limitation in the discussion.

In my previous review I also requested that the manuscript have line and page numbers added, as it make it much easier to review. I was disappointed that the revised version I was sent to review still does not have page and line numbers added.

Some specific comments:

Introduction:

'Data can temper extreme rhetoric..' In this section I would acknowledge that more data is not the answer to all problems. There's studies looking on how organizational biases/personal viewpoints affect selectivity of data and willingness to accept data that disagrees with organizational mission/personal viewpoints. Just one or two sentences need to be added in here to acknowledge that more data does not necessarily lead to better evidence based-policy/decision making - there's more to it than this.

'Such a systems of checks' - be explicit about what is being referred to here

'The services alone develop' - are they really doing this alone? Surely in partnership with interested stakeholders and those with professional expertise.

'Have the greatest understanding' - this still feels a bit weird given that there's not any assessment/evidence that they are making correct determinations at a better rate than other agencies. Maybe, change to say they have the specific mandate to make determinations and have expertise to understand how actions and potential actions affect species.

Box 1 / Fig S1 - The box text needs to start describing the process from an earlier stage. At the moment this text starts with 'Section 7 consultation' when in FWS/NMFS. But in Fig S1 the figure starts with an agency wanting to do an action. The box text needs to start with an agency wanting to do an action, the internal decision within that agency before it goes to FWS/NMFS. How does a action agency make a 'no effect' determination. Then also show what the potential cabinet level exemption process to a jeopardy finding that could not be mitigated would be (in both box 1 and Fig S1). And to confirm, if action agency makes a 'no effect' finding there is no Section 7 process (as

shown in Fig S1). Also, once informal consultation begins in NMFS, I'm still confused what the difference in meaning between that process resulting in a 'NLAA' finding vs a 'no effect' finding is? Please clarify this text.

'Interest in reducing expert agency involvement' - be clear that this paper's scope is in evaluation of how Section 7 process compared between NMFS and action agency. So this is about using the same objective methods/process just about which agency conducts it. This is very different to exemption/reducing/changing the actual evaluation criteria that project is evaluated against.

Results:

Table 1 - I'm confused how NMFS is having disagreements with itself? I guess that this is comparing pre-consultation determinations to post-consultation determinations? This in itself is an interesting exercise, as even the 'experts' can often not get it right without going through the full consultation process. Same for FWS.

'Among species with at least 10 consultations,' - this is confusing as you have the Nassu grouper on the figure with 50% . I assume this is not commented on in the text as too few consultations? I suggest using a fixed threshold of 10 consultations consistently throughout the manuscript.

Discussion:

'It's also possible that NMFS, which manages' - no evidence for this statement in the manuscript. I would suggest removing, or looking at total budget for the consultations between FWS and NMFS and the number of consultations they've had to run to see whether this imbalance really does exist.

'We were unable to systematically assess' - need to more specifically reflect that these results are therefore indicative only. They provide info on the agreement levels between agencies, not which agency made the correct decision and whether even if in agreement the correct decision was made.

'Data/code availability' - Github is not a data archive, as it does not provide a stable static version of this analysis that is permanently archived. Please add data/code used for the final version of this manuscript to a stable archive e.g. dryad, figshare etc and cite with a doi

Fig 1A - add Latin names to the legend for the species

Reviewer 1

The manuscript provides a good commentary on some aspects of the consultation process mandated by the United States Endangered Species Act. The authors introduce the need to examine this topic by referencing proposals to streamline ESA procedures by eliminating consultation by government experts outside the agencies regulating actions that might affect listed species adversely. Thus, their MS is largely in response to political calls to weaken this aspect of ESA implementation.

The authors have addressed many issues raised in previous reviews. They have clarified multiple sections and reduced the emotive language prominent in the first submission. The overarching points regarding the limited nature of the data used to evaluate the consultation process cannot be overcome, and the authors have done a good job extracting what they can from a database that does not contain much in the way of scientific details or biological outcomes of the consultation process.

I remain attentive to the fact that this paper focuses on the minority of cases where administratively defined experts disagreed with the agencies overseeing implementation of the ESA. The paper's findings are pertinent to the administrative aspects of policy implementation, but no assessment of the quality or correctness of the agencies' ultimate determinations or actions is possible at this time.

It remains unclear whether the cases of agreement, as well as those of disagreement, advance or inhibit the task of species recovery. In their response to earlier reviews, the authors note that, due to the lack of data, "it is difficult, if not impossible to make this kind of assessment." Nevertheless, this is the content that many readers will be expecting, given that the title purports to show that "expert wildlife agencies are indispensable to protecting endangered species."

We have added additional clarification in the closing paragraph of the Discussion acknowledging the inability to assess consultation determination correctness with the following sentence:

"We were unable to systematically assess the effectiveness of consultations, or whether NMFS determinations were more correct than those of action agencies, because species status and/or consultation outcome data were lacking. Such data would enable an objective assessment of consultation conclusion accuracy in addition to the comparisons between evaluators presented here."

I continue to feel that this paper is of interest to readers that already understand its implications for the administration of the ESA; others may find it very narrowly focused on a subset of

administrative procedures mandated by the Act. If the authors wish to reach a broader, larger audience, I suggest that they be more direct in describing the contentious political environment in the US, and how current political intervention is putting the Act at risk. This would resonate with international readers, who will then better appreciate the role of expert consultation in the many countries who have modeled their own laws after the US ESA.

As suggested, we have added the following sentence to the opening paragraph of the Introduction more directly describing the political climate in the U.S. and how this threatens the Endangered Species Act:

“In the U.S., a highly-polarized political climate has catalyzed an unprecedented number of legislative proposals to undermine the nation’s conservation laws, often based on the claim that species conservation unduly hinders economic growth.”

Reviewer #2 (Remarks to the Author):

Thank you for responding to my previous review of this manuscript. I recognize that you have addressed many of my previous comments and the manuscript has improved as a result of this. However, I still have some concerns, while many of these are minor, there is still one major concern. Specifically, around the assumption that NMFS determinations are ‘correct’. I appreciate that you are unable to verify this with a third party validation, but I would strengthen the text still further about the staff expertise within NMFS, and then reflect on this limitation in the discussion.

We have added additional sentences in both the Introduction and Discussion to provide additional evidence for NMFS’ unique expertise in endangered species biology, as well as more explicit acknowledgement of our inability to assess the accuracy of consultation outcomes. The exact language added to the manuscript is detailed in our response to specific comments below.

In my previous review I also requested that the manuscript have line and page numbers added, as it make it much easier to review. I was disappointed that the revised version I was sent to review still does not have page and line numbers added.

We have added page numbers to this revision. We were unable to add line numbers, as they prevented successful conversion to PDF by causing 75% of the page to be obscured, both locally and upon uploading to the online submission system.

Some specific comments:

Introduction:

‘Data can temper extreme rhetoric...’ In this section I would acknowledge that more data is not the answer to all problems. There are studies looking on how organizational biases/personal viewpoints affect selectivity of data and willingness to accept data that disagrees with organizational mission/personal viewpoints. Just one or two sentences need to be added in here

to acknowledge that more data does not necessarily lead to better evidence based-policy/decision making - there's more to it than this.

As suggested, we have added the following sentence providing the caveat that factors besides data availability contribute to the use of data in decision making, along with a supporting reference: "Although data availability does not guarantee their use in decision-making⁶, data collection and analysis are the first steps to realizing these benefits."

6. Stephenson, P.J. et al. Unblocking the flow of biodiversity data for decision-making in Africa. *Biol. Conserv.* **213**, 335-340, (2017).

'Such a systems of checks...' - be explicit about what is being referred to here.

This sentence has been edited to state explicitly that the ESA vests expert agencies with the kind of oversight authority described in the previous sentence. It now reads: "...in large part because it vests expert agencies with oversight authority"

'The services alone develop' - are they really doing this alone? Surely in partnership with interested stakeholders and those with professional expertise.

We have revised this sentence to indicate that the Services lead the development of recovery plans. This sentence now reads:

"Because the Services alone have a legal mandate to determine how actions affect listed species, lead the development of plans to recover listed species, and periodically assesses their conservation status, these expert agencies have the greatest expertise in how federal actions affect the species' conservation prospects."

'Have the greatest understanding' - this still feels a bit weird given that there's not any assessment/evidence that they are making correct determinations at a better rate than other agencies. Maybe, change to say they have the specific mandate to make determinations and have expertise to understand how actions and potential actions affect species.

As suggested, we have replaced the term 'understanding' with 'experience' and reference the expert agencies unique mandate to determine how federal actions affect listed species' conservation prospects, as described in the response to the previous comment above.

Box 1 / Fig S1 - The box text needs to start describing the process from an earlier stage. At the moment this text starts with 'Section 7 consultation' when in FWS/NMFS. But in Fig S1 the figure starts with an agency wanting to do an action. The box text needs to start with an agency wanting to do an action, the internal decision within that agency before it goes to FWS/NMFS. How does a action agency make a 'no effect' determination? Then also show what the potential cabinet level exemption process to a jeopardy finding that could not be mitigated would be (in both box 1 and Fig S1). And to confirm, if action agency makes a 'no effect' finding there is no Section 7 process (as shown in Fig S1). Also, once informal consultation begins in NMFS, I'm

still confused what the difference in meaning between that process resulting in a 'NLAA' finding vs a 'no effect' finding is? Please clarify this text.

We have added the following text to the beginning of box 1 describing the internal review done by federal agencies evaluating the effects of their actions on listed species prior to section 7 consultation:

“Federal agencies must determine if actions they plan to take, fund, or permit may affect species on the endangered species list. Agencies can determine on their own authority that a proposed action will have ‘no effect’ on listed species, in which case the action proceeds without involvement of the Services. If an agency determines a proposed action ‘may affect’ listed species...”

Additionally, we clarify in the text that an NLAA finding still indicates that an action may affect a listed species, just not adversely so. This distinguishes NLAA from No Effect findings as follows: “Informal consultation ends if the Services determine that the proposed action will have ‘no effect’ or may affect but is “not likely to adversely affect” (NLAA) a species or critical habitat.”

The cabinet level exemption process is beyond the scope of this paper, as it operates outside of the Services role and ability to conserve imperiled species. We mention it in Box 1 for completeness but have chosen not to include this possible step in Fig S1. Fig. S1 is meant to help clarify the possible combinations of outcomes for action agencies and the Services and including the exemption process would muddle this illustration.

‘Interest in reducing expert agency involvement’ - be clear that this paper’s scope is in evaluation of how Section 7 process compared between NMFS and action agency. So this is about using the same objective methods/process just about which agency conducts it. This is very different to exemption/reducing/changing the actual evaluation criteria that project is evaluated against.

We have changed this sentence to clarify that the proposals we reference do not include exemptions from the evaluation process itself. It now reads:

*“Interest in reducing expert agency involvement continues to this day: exemptions from section consultation **with the Services...**”*

We further clarify this distinction in the following sentence, which has been edited to read:

“A critical, outstanding question is whether self-consultation alternatives effectively conserve species protected by the ESA or simply alleviate conservation obligations.”

Results:

Table 1 - I’m confused how NMFS is having disagreements with itself? I guess that this is comparing pre-consultation determinations to post-consultation determinations? This in itself is an interesting exercise, as even the ‘experts’ can often not get it right without going through the full consultation process. Same for FWS.

We are assuming the comment pertains to Figure 2, not Table 1? Different divisions within NMFS (and FWS) are responsible for creating management actions like fisheries management plans, and conducting section 7 consultations on those actions. We have added text explaining these circumstances to the Figure 2 legend.

‘Among species with at least 10 consultations,’ - this is confusing as you have the Nassau grouper on the figure with 50% . I assume this is not commented on in the text as too few consultations? I suggest using a fixed threshold of 10 consultations consistently throughout the manuscript.

We have revised this figure to include only species and work categories involved in at least ten consultations, consistent with the rest of the manuscript (which eliminates Nassau grouper). The figure legend has been edited accordingly.

Discussion:

‘It’s also possible that NMFS, which manages’ - no evidence for this statement in the manuscript. I would suggest removing, or looking at total budget for the consultations between FWS and NMFS and the number of consultations they’ve had to run to see whether this imbalance really does exist.

As pointed out by the reviewer, we do not provide evidence comparing resource allocation between NMFS and FWS and have removed this part of the statement. We retain the possible explanation that NMFS may have more confidence in its findings as a result of managing fewer species and conducting fewer consultations. This hypothesis is logically plausible, has been suggested by agency personnel, and is supported by a comparison of the number of consultations conducted by each Service.

‘We were unable to systematically assess’ - need to more specifically reflect that these results are therefore indicative only. They provide info on the agreement levels between agencies, not which agency made the correct decision and whether even if in agreement the correct decision was made.

We have added statements in this section to explicitly state that we were unable to assess the accuracy of consultation conclusions, and present this as an important goal of future work. This section now reads:

“We were unable to systematically assess the effectiveness of consultations, or whether NMFS determinations were more correct than those of action agencies, because species status and/or consultation outcome data were lacking. Such data would enable an objective assessment of consultation conclusion accuracy in addition to the comparisons between evaluators presented here.”

‘Data/code availability’ - Github is not a data archive, as it does not provide a stable static version of this analysis that is permanently archived. Please add data/code used for the final version of this manuscript to a stable archive e.g. dryad, figshare etc and cite with a doi

Data and analysis code have been moved to a permanent OSF repository (DOI: 10.17605/OSF.IO/UCFKJ).

Fig 1A - add Latin names to the legend for the species

Latin names for species in Figure 1A are now listed in the legend.